# Characterization of Uterine Motion in Early Gestation Using MRI-Based Motion Tracking

**DOI:** 10.3390/diagnostics10100840

**Published:** 2020-10-19

**Authors:** Thomas Martin, Carla Janzen, Xinzhou Li, Irish Del Rosario, Teresa Chanlaw, Sarah Choi, Tess Armstrong, Rinat Masamed, Holden H. Wu, Sherin U. Devaskar, Kyunghyun Sung

**Affiliations:** 1Radiological Sciences, University of California Los Angeles (UCLA), Los Angeles, CA 90095, USA; thomas.martin@aggiemail.usu.edu (T.M.); XinzhouLi@mednet.ucla.edu (X.L.); TArmstrong1@g.ucla.edu (T.A.); RMasamed@mednet.ucla.edu (R.M.); HoldenWu@mednet.ucla.edu (H.H.W.); 2Department of Obstetrics and Gynecology, University of California Los Angeles (UCLA), Los Angeles, CA 90095, USA; CJanzen@mednet.ucla.edu; 3Department of Epidemiology, UCLA School of Public Health, Los Angeles, CA 90095, USA; IdelRosario@mednet.ucla.edu; 4Department of Pediatrics, University of California Los Angeles (UCLA), Los Angeles, CA 90095, USA; TChanlaw@mednet.ucla.edu (T.C.); choisarahh@gmail.com (S.C.); SDevaskar@mednet.ucla.edu (S.U.D.)

**Keywords:** human pregnancy, placenta MRI, uterine contraction, maternal motion, MRI motion tracking

## Abstract

Magnetic resonance imaging (MRI) is a promising non-invasive imaging technique that can be safely used to study placental development and function. However, studies of the human placenta performed by MRI are limited by uterine motion and motion in the uterus during MRI remains one of the major limiting factors. Here, we aimed to investigate the characterization of uterine activity during MRI in the second trimester of pregnancy using MRI-based motion tracking. In total, 46 pregnant women were scanned twice (first scan between 14 and 18 weeks and second scan between 19 and 24 weeks), and 20 pregnant subjects underwent a single MRI between 14 and 18 weeks GA, resulting in 112 MRI scans. An MRI-based algorithm was used to track uterine motion in the superior-inferior and left-right directions. Uterine contraction and maternal motion cases were separated by the experts, and unpaired Wilcoxon tests were performed within the groups of gestational age (GA), fetal sex, and placental location in terms of the overall intensity measures of the uterine activity. In total, 22.3% of cases had uterine contraction during MRI, which increased from 18.6% at 14–18 weeks to 26.4% at 19–24 weeks GA. The dominant direction of the uterine contraction and maternal motion was the superior to the inferior direction during early gestation.

## 1. Introduction

Placental function and growth are critically important to fetal health and development [1,2,3]. Ultrasound is the primary technique for imaging the fetus and placenta primarily due to its proven utility, easy accessibility, and the low cost compared to other imaging modalities [4]. However, the limitations of ultrasound include low soft-tissue contrast, small field of view (cannot measure placental volume), and poor image quality in some placental functional imaging, which makes ultrasound findings potentially inconclusive or insufficient to guide treatment options [5,6]. MRI is a promising technique for fetal and placental imaging because it provides excellent soft-tissue contrast, functional imaging, and no ionizing radiation [7,8]. However, motion in the uterus during MRI still remains one of the major limiting factors for functional placenta and fetal imaging [9,10] as MRI is sensitive to motion artifacts due to a long acquisition time. To date, there is a limited understanding of the uterine motion, particularly in early gestation, and characterization of the uterine motion would be important to allow developing strategies to minimize motion-related MRI artifacts.

The motion of the placenta and uterus during MRI can be caused by multiple intra- and extra-uterine factors. Intra-uterine factors include uterine contractions and fetal motion while extra-uterine factors include maternal respiration and other organ motion, such as bowel peristalsis [11,12]. In particular, the uterine contractions can compress the superior region of the uterus, causing significant motion in the uterus and placenta. Dickinson et al. studied the uterine contraction activity during gestational ages (GAs) of 20–40 weeks and showed that there is a mean contraction frequency in pregnant women of ~2 contractions/h [13]. Although the study provided the frequency and duration of the uterine contraction, the contraction measurements were not conducted during MRI, which is generally less than 5 min for each MRI sequence. Therefore, an MRI scan acquired with an imaging sequence of approximately 5 min would be adequate to characterize the frequency, duration, and directionality of the uterine contraction.

A recent study using electrocardiogram (ECG) for long-term monitoring of the fetus and mother while at home revealed uterine contractions are associated with a graded response in fetal heart rate and may represent a physiological challenge for the development and adaptation of the fetal cardiovascular system [14]. Accurate detection of uterine contractions may provide valuable clinical information on fetal development early in gestation. The study used monitors recording electrophysiological signals of the fetal heart electrocardiogram (ECG) and uterine contractions by the electro-hysterogram (EHG). However, only about 70% of the time was the fetal heart rate accurately measured, and the EHG reported a 19% false positive detection of uterine contractions. Using MRI one can potentially reduce the false positive detection of uterine contractions, by direct visualization of uterine and fetal activity.

In this study, we report and characterize uterine activity during MRI in women with normal pregnancies as part of the Human Placenta Project [15]. We implemented an MRI-based motion tracking algorithm [16] to generate the uterine motion plot, providing information about the uterine activity in the superior/inferior (SI) and left/right (LR) axes. In normal pregnancies, uterine contractions and motion caused by maternal motion (e.g., maternal respiration and other maternal organ motion), separated by an abdominal radiologist and a maternal-fetal medicine specialist, were characterized by criteria based on their intensity and direction dependent on the uterine motion plot. Associations with GA, fetal sex (female vs. male), and placental position (anterior vs. posterior) were investigated in the cases of uterine contraction and maternal motion in a total of 112 MRI scans.

## 2. Materials and Methods

### 2.1. Study Population

The UCLA Institutional Review Board (IRB) approved this prospective study (IRB #15-001388, approved on 2 October 2015). All subjects provided written informed consent, and the study was compliant with the Health Insurance Portability and Accountability Act. Without pre-selection, we screened for all eligible pregnant women at the UCLA prenatal clinic. The eligibility criteria included subjects less than 14 weeks GA, more than 18 years old, viable pregnancy, not carrying twins, and planning to deliver at the same institution. We retrospectively reviewed the subjects with placental insufficiency, confirmed by clinical information inclusive of maternal and neonatal outcomes at the time of birth, and excluded these pregnancies from the study to capture normal pregnancies alone without complications. A total of 66 normal pregnant women (mean age: 35 ± 5 years, range: 22–45 years) were included in the study. Forty-six of the subjects were scanned twice at two different GA during their second trimester, with the first scan (1st) between 14 and 18 weeks, and the second scan (2nd) between 19 and 24 weeks. The other 20 subjects underwent only their first MRI (14–18 weeks GA) and were lost to the follow up second scan. In total, there were 112 MRI scans performed.

### 2.2. MRI Techniques and Reconstruction

All MRI scans were performed at 3T (Prisma or Skyra, Siemens Healthcare, Erlangen, Germany) using a body array coil. Subjects were positioned feet-first supine. Physiological monitoring was not required due to scans being performed during early GA. An anatomical T_2_-weighted scan of the abdomen/pelvic region was performed using a T_2_-HASTE sequence. This was used to identify the placenta, uterus, and other relevant anatomical structures. A 3-D multi-echo golden-angle-ordered radial gradient echo (GRE) sequence (3–6 min) was used for the assessment of uterine contractions and maternal motion. The detailed MRI scan parameters are listed in Table 1. A dynamic set of images were retrospectively reconstructed offline using the k-space weighted image contrast (KWIC) technique [17]. Only the first echo was used for reconstruction because of the best qualitative signal of the placenta, uterus, and fetus. We used 10 radial spokes in the center of k-space and 170 spokes in the outer most ring, resulting in an apparent temporal resolution of 4.2 to 7.6 s after the KWIC reconstruction. The dynamic images were reconstructed in axial slices and reformatted to coronal images. The coronal and axial images were used to determine uterine and maternal motion in the SI and LR directions, respectively.

### 2.3. Image-Based Template-Matching Motion Tracking

We used an in-house software developed for tracking uterine motion employing an image-based template matching algorithm [16]. The uterine motion along the SI and LR directions was measured on 2-D coronal and axial images, which covered the uterus, reformatted from the 3-D dynamic image sets, respectively (Figure 1 and Appendix A shows the movie). The inner rectangle (dashed-line) depicts the target region-of-interest (ROI), and the outer rectangle (solid line) delineates the search region to ensure a local target matching on the tracking images based on expected ranges of motion. The initial target ROI was manually defined on the image in the first-time frame to cover the superior or left/right regions of the uterus. A least-squares-based comparison metric was used to optimize the matching position of the target feature at each dynamic frame of the MR images. The initial target ROI position was then subtracted from the best-matched target ROI location in the search region to calculate motion coordinates in two directions in each time frame.

The center-of-mass of the inner rectangle location (x-y coordinate) was determined for each time point. The y-coordinate of the center-of-mass tracking represented the SI direction of the motion on coronal images, and the x-coordinate of the center-of-mass tracking represented the LR direction of motion on axial images. We then performed motion tracking on multiple coronal and axial slices within the uterus. The center-of-mass location was averaged across all slices within the uterus for each time point to account for any random fluctuations that may occur while tracking uterine motion. The mean center-of-mass of the inner rectangle was determined in units of mm for each time point, resulting in the motion time plot for SI and LR directions. We then combined these two plots by taking a square root of the sum of two squares to generate the uterine motion time plot, *U*(*t*). Figure 2 illustrates the motion time plots for SI (blue) and LR (green) directions (Figure 2a,c) and their corresponding uterine motion time plots in red (Figure 2b,d).

### 2.4. Uterine Contraction and Maternal Motion

An experienced abdominal radiologist (R.M.; with 10+ years of experience) and an experienced maternal-fetal medicine specialist (C.J.; with 20+ years of experience) individually reviewed the dynamic MR images and confirmed the uterine contraction based on the anatomical identification of the uterus and uterine myometrium versus fetus, maternal bowel, and diaphragm. These structures were well-visualized in the MR images, and all the uterine contraction cases were matched between two expert readers. Once the uterine contraction was confirmed, the frequency of the uterine contractions was recorded by the number of the MRI scans with uterine contractions divided by the total number of MRI scans. The intensity of the uterine contraction was quantified by the maximum amplitude, *U_max_* (mm), and the normalized area under the curve, *AUC_norm_* (mm), of *U*(*t*). *AUC_norm_* was defined as:
(1)$$AUC_{norm}=\frac{1}{T_{GRE}}\int_{\tau \in \Omega }U\left(\tau \right)d\tau ,$$
where *T_GRE_* is the imaging time (s) for 3-D multi-echo golden-angle-ordered radial GRE, and Ω is the time duration of the uterine contraction, defined by the shift greater than 1 mm in *U*(*t*). An example of *U*(*t*) with the uterine contraction is shown in Figure 2b. The AUC of the uterine contraction is illustrated as the shaded areas. Other motion that was not part of the uterine contraction was attributed to maternal motion, caused by extrauterine movement (respiratory, subject, and other organ motion) or non-contraction motion. The amount of maternal motion was quantified by *U_max_*. An example of *U*(*t*) with the maternal motion is shown in Figure 2d. Additionally, the direction of the uterine motion, θ, was calculated at *U_max_*, defined by 90° being SI only and 0° being LR only. The uterine motion was determined to be SI dominant when θ was higher than 45°.

### 2.5. Statistical Analysis

Data are presented as mean ± standard deviation and stratified in three ways: (i) GA (1st and 2nd MRI), (ii) fetal sex (male and female), and (iii) placental location (anterior and posterior) [18]. The baseline characteristics of GA and the intensity of uterine motion were tested within the groups of GA, fetal sex, and placental location using the unpaired Wilcoxon tests. *p*-values less than 0.05 were considered statistically significant, and less than 0.01 were considered highly significant.

## 3. Results

### 3.1. Uterine Contraction

The baseline characteristics of GA showed a significant difference between the first and second MRI scans (*p* < 0.001) and no difference between groups based on fetal sex or placenta location (Table 2). In total, 22.3% (25 out of 112) of the MRI scans had uterine contractions at 19.0 ± 2.6 weeks GA, where all the uterine contractions were identified independently by both clinicians with varied expertise in imaging, namely radiology or maternal-fetal medicine (R.M. and C.J). We observed an increase in contraction frequency with advancing gestation. The uterine contractions occurred in 18.6% of the MRI scans during 14–18 weeks GA and 26.5% of the MRI scans during 19–24 weeks GA. The frequency of uterine contractions was similar between male and female fetal sex while there is a relative increase in posterior (25%) compared to anterior (20%) placentas.

We visualized patterns of uterine contractions by combining the intensity measure (*AUC_norm_*) and direction (*θ*) in polar coordinates. Figure 3 includes all uterine contractions (a), seen in first MRI (b) and second MRI (c) scans. Both intensity measures of uterine contractions (*AUC_norm_* and *U_max_*) were not significantly different between first and second MRI scans (*p* = 0.76 and 0.46), and 84% of the uterine contraction cases were located in *θ* > 45°, suggesting the SI direction to be the dominant direction. The uterine contractions between male and female fetal sex and between anterior and posterior placentas are shown in Figure 4. Both *AUC_norm_* and *U_max_* were not different between the two fetal sexes or placental positions, with SI being the dominant direction of uterine contractions observed consistently (Table 3).

### 3.2. Maternal Motion

Patterns of maternal motion were visualized by combining the maximum displacement (*U_max_*) and direction (*θ*) in polar coordinates (Figure 5). The scatter plots pertaining to the polar coordinates displayed the dominant direction to be SI in most cases (67.8% of all cases). The unpaired Wilcoxon test revealed no significant difference in the maximum displacement, *U_max_*, between the two temporally acquired MRI scans, fetal sex, or placental position (Table 4).

## 4. Discussion

MRI is a promising non-invasive imaging technique that can provide important placental structural and functional information in early gestation [8], but intra-uterine and extra-uterine motion during MRI scanning is one of the major limiting factors. We used an MRI-based template-matching algorithm to characterize uterine contractions separate from maternal motion in normal human pregnancy during the second trimester. The uterine contraction activity was characterized by the frequency, intensity, and direction during MRI (*n* = 112). About 22% of cases exhibited uterine contractions during MRI acquisition, which were 3–6 min in duration, with a dominance of the superior to inferior direction in about 84% of the cases where uterine contractions were detected. Therefore, motion compensation strategies along the SI direction should be recommended for placental MRI studies.

Dickinson et al. [13] performed a study assessing uterine contractions, which involved monitoring the frequency, duration, and amplitude of contractions in low-risk pregnant women at gestational ages between 20 and 40 weeks using ambulatory tocodynamometry. In 20 women, they observed an increase in the frequency of uterine contractions with advancing gestation. Our present findings (from 18.6% at 16.5 weeks to 26.4% at 20.9 weeks) obtained from 112 MRI scans are consistent with these previous results. Our study observed that the intensity of the uterine activity was not associated with differences in fetal sex, placental location, and GA during the second trimester; however, these associations may change in the third trimester as described by Dickinson et al., where substantial alterations in the contractile profiles were observed after 29 weeks GA [13].

We also observed that there was no significant difference in the intensity of maternal motion based on GA in the second trimester, fetal sex, and placental location (*p* > 0.05). The intensity of extrauterine motion (e.g., maternal respiration and other maternal organ motion) was relatively smaller than that of uterine contractions and had no association with GA, fetal sex, or placental location. This observation supports the existent notion that motion due to extraneous maternal causes is spontaneous and random; thus, we speculate that these characteristics pertinent to maternal motion would be no different at later gestational periods.

The 3-D multi-echo golden-angle-ordered radial GRE sequence that we used can provide more than motion characterization alone. A recent study showed that this sequence can be used for free-breathing fat quantification and R_2_* measurements [19]. Though yet to be clearly understood, future work may decipher if there indeed is placental fat and whether such quantification could prove to be useful in predicting the impact of the maternal metabolic state on placental health and pregnancy outcomes. Since we collected multiple echoes, it is possible to generate R_2_* maps, thereby providing information on placental oxygenation. Using the motion information generated in this study, one can strive to reduce motion artifacts in interpreting scans that demonstrate significant motion.

One of the limits of our study is the limited number of uterine contractions noted in scans. Even though the posterior placentas were associated with a trend towards a higher rate of uterine contraction when compared to anterior placentas, future studies with a larger sample size may provide the power necessary to detect significant differences. Nonetheless, we successfully recruited 66 normal pregnant women and included 112 MRI scans (3–6 min) during the second trimester to characterize the frequency of uterine contractions. Another challenge is the low image contrast provided between the placenta, fetus, and other features in the 3-D GRE sequence. This can make it challenging to distinguish between the causes of uterine motion due to fetal motion, maternal digestive track, or uterine contractions. We attempted elimination of possible ambiguity due to the different causes contributing to uterine motion. However, in the future, other MRI sequences may be employed towards further improving the contrast images for motion tracking, such as T_2_-weighted or T_2_/T_1_-weighted image contrast (e.g., spin echo or balanced steady-state free precession). However, the trade-off to these aforementioned techniques is slower image acquisition, banding artifacts, and/or a decreased number of slices. Thus, there is no flawless technique so far that quantifies the differing reasons for uterine motion.

## 5. Conclusions

We investigated uterine motion during MRI acquisition in normal human pregnancy at early gestation, characterized by MRI-based motion tracking. We showed that there is an increase in uterine contraction frequency between 14 and 18 weeks (18.6%) and 19 and 24 weeks (26.4%) GA while the dominant direction of the contraction was consistently in the superior to inferior direction. There were no significant differences in the intensity of observed uterine activities based on GA, fetal sex, or placental position during early gestation as in the second trimester. This characterization of uterine motion will prove to be of significant importance in developing functional placental and fetal MRI acquisition strategies that appropriately compensate for and/or minimize motion artifacts, which would be a key factor to help gain a better understanding of the normal physiology of pregnancy. In addition, compilation of uterine contractions separate from other extraneous maternal/fetal causes of motion may provide the ability to study them during various states of pregnancy-associated complications in the future.

## Figures and Tables

**Figure 1 diagnostics-10-00840-f001:**
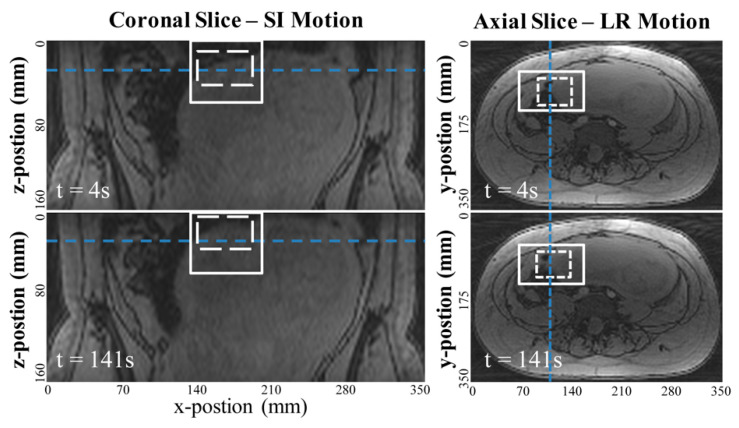
Example images measuring the uterine motion in mm with the image-based template matching algorithm. The coronal reformatted images were used to measure superior-inferior (SI) motion and axial images were used to measure left-right (LR) motion. The template matching algorithm searches for the target ROI (dotted white square) within the search region (solid squares) for each dynamic time frame, using a least-squares comparison metric based on normalized image intensity.

**Figure 2 diagnostics-10-00840-f002:**
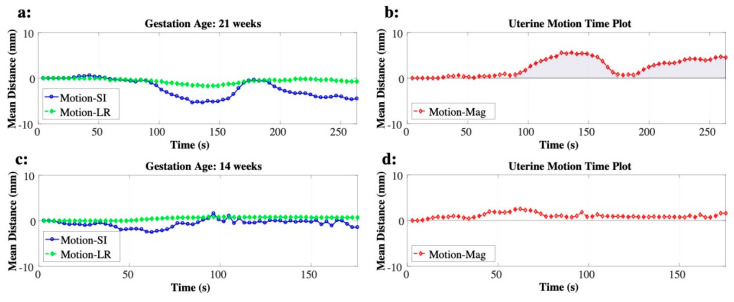
Examples of uterine motion time plots obtained by combining the two motion plots for superior-inferior (blue) and left-right (green) directions. Examples of uterine contraction (**a**,**b**) and of maternal motion (**c**,**d**) are shown. The area under the curve of the uterine contraction is shown as the shaded area.

**Figure 3 diagnostics-10-00840-f003:**
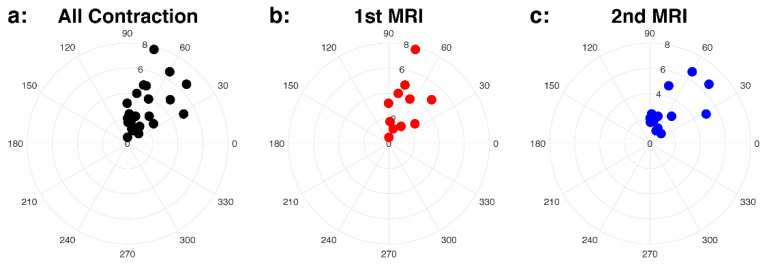
Intensity (*AUC_norm_*) and direction (*θ*) of the uterine contraction in the polar coordinates for all cases (**a**), first (1st) MRI, colored as red, (**b**), and second (2nd) MRI, colored as blue, (**c**) cases. The polar plots show the dominant direction is SI.

**Figure 4 diagnostics-10-00840-f004:**
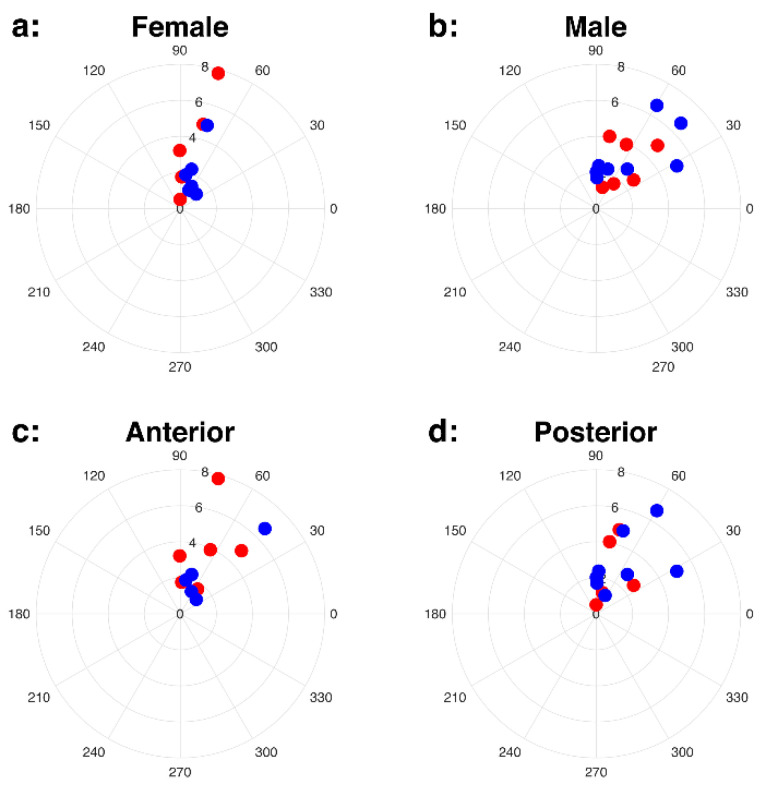
Uterine contraction for female (**a**) and male (**b**) fetal sex, and anterior (**c**) and posterior (**d**) placentas. First (1st) and second (2nd) MRI scans are colored as red (1st MRI) and blue (2nd MRI). The scatter plots in the polar coordinates show that the dominant direction of motion is SI.

**Figure 5 diagnostics-10-00840-f005:**
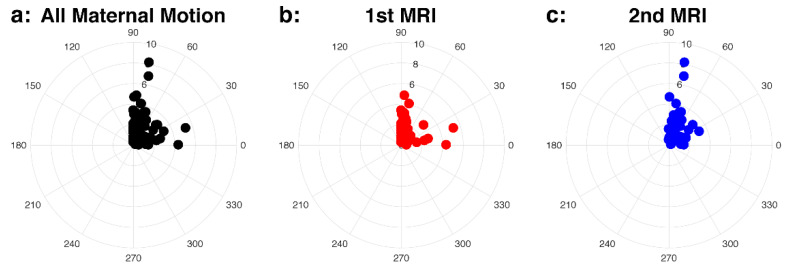
Intensity (*U_max_*) and direction (*θ*) of the maternal motion in the polar coordinates for all cases (**a**), first (1st) MRI (**b**), and second (2nd) MRI (**c**) cases.

**Table 1 diagnostics-10-00840-t001:** Sequence scan parameters for in vivo MRI scans.

Imaging Parameters	3-D Multi-Echo GA Radial GRE
Number of Echoes	6–12
TE_1_, ms	1.23
ΔTE, ms	1.23
TR, ms	8.85–15.9
In-plane Resolution, mm × mm	1.7–2.2 × 1.7–2.2
FOV, mm	350–380
Slice Thickness, mm	3.5–5
Flip Angle, degrees	5
Scan Time, min: sec	3:18–6:06

**Table 2 diagnostics-10-00840-t002:** The baseline characteristics of the gestational age (GA) within the groups of GA, fetal sex, and placental location.

Groups		Uterine Contraction		Number of Contraction/Totals (%)	Maternal Motion	
		GA, Weeks	*p*-Value		GA, Weeks	*p*-Value
All Cases		19.0 ± 2.6		25/112 (22.3)	17.8 ± 2.6	
Longitudinal	1st MRI	16.5 ± 1.0	<0.001 **	11/59 (18.6)	15.8 ± 1.0	<0.001 **
	2nd MRI	20.9 ± 1.4		14/53 (26.4)	20.4 ± 1.2	
Fetal Sex	Female	18.8 ± 2.4	1	11/50 (22.0)	18.0 ± 2.6	0.57
	Male	19.1 ± 2.8		14/62 (22.6)	17.7 ± 2.5	
Placental Position	Anterior	18.9 ± 2.3	0.87	12/60 (20.0)	18.0 ± 2.6	0.51
	Posterior	19.0 ± 2.9		13/52 (25.0)	17.7 ± 2.5	

** Highly significant (*p* < 0.01).

**Table 3 diagnostics-10-00840-t003:** The motion characteristics (intensity and direction) of the uterine contraction within the groups of the gestational age (GA), fetal sex, and placental location.

Groups		AUC_norm_, mm	*p*-Value	U_max_, mm	*p*-Value	*θ*, Degrees	Number of SI Dominant/Totals (%)
All Cases		3.1 ± 2.0		7.8 ± 5.5		66.7 ± 18.1	21/25 (84)
Longitudinal	1st MRI	3.3 ± 2.1	0.76	6.7 ± 4.3	0.46	69.3 ± 17.7	10/11 (90.9)
	2nd MRI	3.0 ± 1.9		8.7 ± 6.3		64.7 ± 18.9	11/14 (78.6)
Fetal Sex	Female	2.8 ± 2.2	0.20	7.4 ± 4.2	0.94	72.6 ± 14.4	10/11 (90.9)
	Male	3.4 ± 1.8		8.1 ± 6.4		62.0 ± 19.9	11/14 (78.6)
Placental Position	Anterior	3.2 ± 2.2	0.98	8.0 ± 4.8	0.61	65.4 ± 16.7	10/12 (83.3)
	Posterior	3.0 ± 1.9		7.6 ± 6.2		67.9 ± 19.9	11/13 (84.6)

**Table 4 diagnostics-10-00840-t004:** The motion characteristics (intensity and direction) of the maternal motion within the groups of the gestational age (GA), fetal sex, and placental location.

Groups		U_max_, mm	*p*-Value	*θ*, Degrees	Number of SI Dominant/Totals (%)
All Cases		2.1 ± 2.1		60.0 ± 29.9	59/87 (67.8)
Longitudinal	1st MRI	1.8 ± 1.2	0.48	65.7 ± 29.9	35/48 (72.9)
	2nd MRI	2.4 ± 2.8		52.9 ± 28.7	24/39 (61.5)
Fetal Sex	Female	2.2 ± 2.7	0.77	61.2 ± 29.2	27/39 (69.2)
	Male	2.0 ± 1.4		59.0 ± 30.7	32/48 (66.7)
Placental Position	Anterior	2.0 ± 1.5	0.50	58.0 ± 30.3	30/48 (62.5)
	Posterior	2.1 ± 2.6		62.3 ± 29.5	29/39 (74.3)

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
