# Peer review of "Characterization of Uterine Motion in Early Gestation Using MRI-Based Motion Tracking"

_diagnostics, 2020, doi:10.3390/diagnostics10100840_

Round 1

Reviewer 1 Report

Martin et al characterize uterine activity during MRI in normal pregnancy, using a motion tracking algorithm to generate a uterine motion plot. The study utilized previously collected sequences.

The data provide information on the number of contractions that typically occur during a MRI – which may not be representative of pregnancy as it is not routine to have an MRI in pregnancy.

The authors state a major premise is that by defining and knowing the baseline rate of uterine activity will be useful for motion correction strategies for prenatal diagnosis and placental evaluation by MR. However, sequences that last 3-6 minutes, such as their gradient echo sequence, is not the standard for characterization of pathology of the uterus. Typically fetal imaging (evaluation of uterine pathology) is performed with single shot fast spin echo (SSFSE), thus motion artifact correction is not necessary.

That said, functional imaging (eg ASL), does rely on longer acquisitions and their findings may be useful in that situation. The authors should modify their statements of the role for motion collection – that it is for the long acquisitions – not for fetal imaging or characterization of uterine pathology.

The authors determined if the activity was due to uterine contractions vs non uterine motion (maternal breathing, bowel peristalsis, etc) through evaluation of two experts. It is not clear how the experts made these determinations and how reliable they are.

Finally, the paragraph in the introduction related to the “well known that uterine contractions pose a physiologic challenge for the fetal cardiovascular system” should be toned down. Uterine contractions are a normal physiologic process and the fetus is well positioned to handle them. Given that the authors did not evaluate the impact of the uterine contractions in this study, I would delete this from the introduction.

Author Response

The authors state a major premise is that by defining and knowing the baseline rate of uterine activity will be useful for motion correction strategies for prenatal diagnosis and placental evaluation by MR. However, sequences that last 3-6 minutes, such as their gradient echo sequence, is not the standard for characterization of pathology of the uterus. Typically fetal imaging (evaluation of uterine pathology) is performed with single shot fast spin echo (SSFSE), thus motion artifact correction is not necessary. That said, functional imaging (eg ASL), does rely on longer acquisitions and their findings may be useful in that situation. The authors should modify their statements of the role for motion collection – that it is for the long acquisitions – not for fetal imaging or characterization of uterine pathology.

Thanks for the valuable comment. We modified the text to properly address the potential benefits of understanding uterine contraction and other motion characteristics (Page 1, Line 44-45).

The authors determined if the activity was due to uterine contractions vs non uterine motion (maternal breathing, bowel peristalsis, etc) through evaluation of two experts. It is not clear how the experts made these determinations and how reliable they are.

The uterine contraction was confirmed based on the anatomical identification of the uterus and uterine myometrium vs. fetus, maternal bowel, and diaphragm, and all the uterine contraction cases were matched between two readers. This is now clarified in the text (Page 4, Line 151-154).

Finally, the paragraph in the introduction related to the “well known that uterine contractions pose a physiologic challenge for the fetal cardiovascular system” should be toned down. Uterine contractions are a normal physiologic process and the fetus is well positioned to handle them. Given that the authors did not evaluate the impact of the uterine contractions in this study, I would delete this from the introduction.

Thanks for pointing out, and we have removed the sentence (Page 2, Line 59).

Reviewer 2 Report

The manuscript is well-written and illustrated. I recommend this paper for publication.

Author Response

We appreciate your comment, and thanks a lot for reviewing our manuscript.

Reviewer 3 Report

In their prospective research article, Martin et co-workers aimed to investigate the characterization of uterine activity during MRI in second trimester pregnancy using MRI-based motion tracking in normal pregnancies. They included 46 pregnant women, comparing those scanned twice (between 14-18 weeks and 19-24 weeks) with those underwent a single MRI between 14-18 weeks GA. In a total of in 112 MRI scans analyzed by an in-house software developed for tracking uterine motion employing an image-based template matching algorithm, they found that 22.3% of cases had 29 uterine contraction during MRI, which increased from 18.6% at 14-18 weeks to 26.4% at 19-24 weeks GA. The dominant direction of the uterine contraction and maternal motion was the superior to inferior direction during early gestation.

The topic is of fascinating interest, with a real impact on the knowledge of a non-invasive imaging technique more and more useful in feto-maternal medicine, ranging from placental disorders to fetal abnormalities. In the scenario of assessing strategies to minimize motion-related MRI artifact, this study is promising. The paper is well-written, with a good readability. The methodology is robust, detailed and reproducible. The study design appropriate, with a posteriori exclusion of abnormal cases in order to endorse the interpretation of the results. The results are detailed adequately. The discussion includes comparison with available evidence until now and opens to future research directions. In the opinion of this reviewer, the paper deserves to be published.

Author Response

(The authors gave the same response as above.)

Round 2

Reviewer 1 Report

There remain statements (as one example see conclusion paragraph) that this is "a key factor to improve MRI as an imaging modality for prenatal diagnosis" - as stated in review this is not the case. Most fetal imaging for prenatal diagnosis are rapid scans that do not require motion correction.

They need to do more to address the issues in point 1 that are permeated through the manuscript - eg in conclusion....

Author Response

Thanks for pointing out an important aspect of the paper. We have rephrased the Conclusion accordingly (Page 9, Line 278-280).